# Systematic review of changed smoking behaviour, smoking cessation and psychological states of smokers according to cigarette type during the COVID-19 pandemic

Hae-ryoung Chun  ,[1] Eunsil Cheon,[1] Ji-eun Hwang  [2]

[1]Graduate School of Public Health, Seoul National University, Gwanak-gu, Seoul, Republic of Korea
[2]College of health science, Dankook University, Chungnam, Republic of Korea

**Correspondence to**
Dr Ji-eun Hwang;
loshjeve@gmail.com

## ABSTRACT

**Objectives** Although the global COVID-19 pandemic has increased interest in research involving high-risk smokers, studies examining changed smoking behaviours, cessation intentions and associated psychological states among smokers are still scarce. This study aimed to systematically review the literature related to this subject.

**Design** A systematic review of published articles on cigarettes and *COVID-19*-related topics

**Data sources** Our search was conducted in January 2021. We used the keywords COVID-19, cigarettes, electronic cigarettes (e-cigarettes) and psychological factors in PubMed and ScienceDirect and found papers published between January and December 2020.

**Data selection** We included articles in full text, written in English, and that surveyed adults. The topics included smoking behaviour, smoking cessation, psychological state of smokers and COVID-19-related topics.

**Data extraction and synthesis** Papers of low quality, based on quality assessment, were excluded. Thirteen papers were related to smoking behaviour, nine papers were related to smoking cessation and four papers were related to psychological states of smokers.

**Results** Owing to the COVID-19 lockdown, cigarette users were habituated to purchasing large quantities of cigarettes in advance. Additionally, cigarette-only users increased their attempts and willingness to quit smoking, compared with e-cigarette-only users.

**Conclusions** Owing to the COVID-19 outbreak, the intention to quit smoking was different among smokers, according to cigarette type (cigarette-only users, e-cigarette-only users and dual users). With the ongoing COVID-19 pandemic, policies and campaigns to increase smoking cessation intentions and attempts to quit smoking among smokers at high risk of COVID-19 should be implemented. Additionally, e-cigarette-only users with poor health-seeking behaviour require interventions to increase the intention to quit smoking.

## INTRODUCTION

On 30 January 2020, the WHO declared COVID-19, an infectious disease of the respiratory tract characterised by SARS,[1] a public health emergency of international concern.[2]

## STRENGTHS AND LIMITATIONS OF THIS STUDY

⇒ This study is the first to systematically review the literature regarding changes in smoking behaviours, cessation intentions and psychological states of smokers, based on the study period, country and age of participants during the COVID-19 pandemic.

⇒ According to the Risk of Bias Assessment Tool for Non-randomised Studies, three researchers cross-checked and evaluated the articles' quality and excluded papers with a high risk of bias.

⇒ Using the backward snowballing method, screening for article selection was strictly performed by checking for any additional papers.

⇒ Since the articles collected for this systematic review were only cross-sectional designs, it was difficult to determine causality; therefore, only associations were presented.

In 2020, a total of 350 594 336 cases of COVID-19 were reported, including 7 893 839 deaths.[3 4] Due to the pandemic, different standards and steps to control the spread of the disease have been adopted worldwide, but almost all countries have implemented lockdown and social distancing measures to prevent COVID-19 infection.

Smoking is a high-risk factor for COVID-19 infection, and many systematic reviews and meta-analyses report the association between COVID-19 and smoking.[5 6] Since tobacco is associated with chronic lung disease, physicians could expect more cigarette users to develop a severe form of COVID-19,[5] given the effect of tobacco on respiratory disease and immune function.[7] Smoking is associated with negative progression and side effects of COVID-19.[6] Smoking, including current or previous smoking history, significantly increases the severity and risk of death from COVID-19.[8] Similar to cigarette use, e-cigarette use can cause oxidative stress and

inflammatory response in the lungs, making e-cigarette users more susceptible to bacterial or viral infections.[9] Although no literature indicates that cigarettes are the direct cause of COVID-19 infection, smokers have a higher risk of severe COVID-19 symptoms than non-smokers.[10] Previous studies have suggested several public health messages to continue focusing on smoking cessation efforts; however, smoking rates have not decreased.[11]

Few investigations have focused on whether smoking behavior[12] and cessation efforts[13] have changed during the COVID-19 pandemic. Previous studies have shown that smoking has decreased[12] and quitting attempts have increased[13] during the pandemic. Since lockdowns and various social distancing measures were implemented in the early stages of the COVID-19 pandemic, they have the potential to affect smoking behaviour and cessation intention. In some countries, it is expected that smoking behaviour will change given the difficulty in freely purchasing tobacco products as non-essential items; individuals are locked down under the pandemic and are required to stay at home for prolonged periods.[14] Changes in daily life due to social distancing and lockdowns are expected to help reduce smoking consumption.[15] Therefore, it is necessary to examine the factors influencing smoking behaviour and cessation intention in the early stages of the pandemic, which have been investigated in the literature.

There are insufficient studies on smokers' negative psychological states during the COVID-19 pandemic. Further research is needed because the relationship between negative psychological states and health behaviour is unclear.[16] People experience higher stress levels than usual because of social isolation, employment insecurity, finances uncertainty, responsibilities and concerns about illness from the virus.[17] Therefore, many cigarette users smoke to relieve stress and negative emotions,[18] but it is often counterproductive.[19] Previous research has shown that negative psychological states, such as stress, anxiety and depression, can influence tobacco-use behaviour.[20]

There are insufficient studies to compare the smoking behaviour, cessation intention and psychological state of cigarette and e-cigarette users during the COVID-19 pandemic.[14] The daily use of e-cigarettes among e-cigarette users slightly increased during the COVID-19 lockdown[21]; therefore, it is necessary to investigate whether smoking behaviour changes according to the cigarette type. Cigarette users are aware that they engage in harmful health behaviours. However, it is unclear whether e-cigarettes help cigarette users quit smoking.[22] E-cigarette users believe that e-cigarettes are less harmful to their health as they contain relatively fewer harmful ingredients than cigarettes.[23] Therefore, they believe that smoking e-cigarettes is an effort to quit smoking.[23]

In addition, after the COVID-19 pandemic, smoking behaviour may change by age depending on the cigarette type. The use of e-cigarettes increases among persons in their early 20s compared with old persons.[24] In addition,

it is known that older persons are more vulnerable to COVID-19 infection and can develop severe infection symptoms.[25] Therefore, there may be an increase in the older persons' intention to quit smoking cigarettes. However, young people who use e-cigarettes may not be less willing to quit smoking.

Many studies have investigated whether smoking is a high-risk factor for COVID-19.[6 8] However, relatively few studies have been published on whether smoking behaviour, cessation intention and psychological state have changed during the COVID-19 pandemic. In the future, it is expected that this study will establish a scientific basis and policy response to smoking behaviour, cessation intention and psychological state based on cigarette type, both during and post COVID-19. Therefore, we conducted a systematic review based on the hypothesis that there would be changes in smoking behaviour, cessation intention and psychological states of adult smokers, depending on the tobacco product used during the COVID-19 pandemic.

## METHODS
### Literature search strategy
We systematically reviewed the data according to the Preferred Reporting Items for Systematic Reviews and Meta-Analyses, shown in the Supplementary file. We searched articles using MeSH terms 'COVID-19, depression, stress, psychological, anxiety, tobacco, tobacco products, tobacco use and electronic nicotine delivery system' in January 2021. Only papers published between January and December 2020 were included in the study. All search terms are listed in online supplemental table 1.

A literature search was conducted on PubMed using the following MeSH terms: '(depression OR stress OR anxiety OR psychology) AND ((tobacco OR tobacco products OR tobacco use OR electronic nicotine delivery system) AND COVID-19))' in the title or the abstract in 2020. A literature search was conducted on ScienceDirect using the following search terms: 'COVID-19 AND (e-cigarette OR "electronic cigarette" OR "electronic nicotine delivery" OR "vaping" OR "heat not burn" OR "heated tobacco product")', as well as COVID-19 AND (tobaccos OR cigarette OR kretek OR bidis OR "pipe tobacco" OR cigarillos).

### Inclusion and exclusion criteria
Figure 1 shows a flowchart of the database searches, followed by the exclusion/inclusion strategy. The inclusion criteria were cases that did not meet the exclusion criteria. In the first phase (figure 1), certain types of articles were excluded, including comments, letters, editorials, viewpoints, correspondence and articles without full text. In addition, duplicate works and those not written in English and studies that did not include humans or adults were excluded. In the next phase (figure 1), studies unrelated to tobacco or COVID-19 were excluded. In addition, articles were excluded if the topics were not related

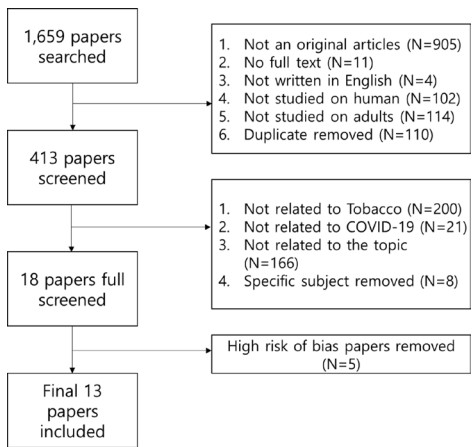

**Figure 1** Flowchart diagram. The flowchart shows the article selection process, the criteria for exclusion of articles and the number of articles excluded.

to changes in tobacco use and mental health caused by COVID-19. Finally, articles with a high risk of bias were excluded using the Risk of Bias Assessment Tool for Non-randomised Studies (RoBANS) (figures 2 and 3).[26]

### Data extraction

Two independent reviewers (H-rC and J-eH) screened the literature and assessed each paper by reading the titles, abstracts and full texts. Three authors extracted data from studies that fulfilled our inclusion criteria independently (H-rC, J-eH and EC). The primary data points included

were the following: study details (author, journal, publication date, country, study design, study period and funding), the total number of participants, types of cigarettes, changes in smoking behaviour, cessation intention and psychological state.

Two authors (H-rC and EC) independently assessed the quality of included studies using the RoBANS.[26] If the opinions of the two authors were different, the quality evaluation was completed through further discussion with another researcher (J-eH). The RoBANS contains six domains: selection of participants, confounding variables, measurement of exposure, blinding of outcome assessments, incomplete outcome data and selective outcome reporting (figures 2 and 3). The risk of bias was divided into three stages (high, unclear or low risk). When the risk of bias in a study was high, it was excluded.

Using the backward snowballing method,[27 28] we checked the references of the articles that met the inclusion criteria. We checked the titles, journals, abstracts and full texts of the references of the articles. We excluded articles that did not meet the inclusion criteria. In addition, articles not related to the subject of this study were excluded. As a result, no additional papers were added.

### RESULTS

#### Literature search and literature selection results

A total of 1659 papers were found in PubMed and Science-Direct, and 905 non-original papers were excluded.

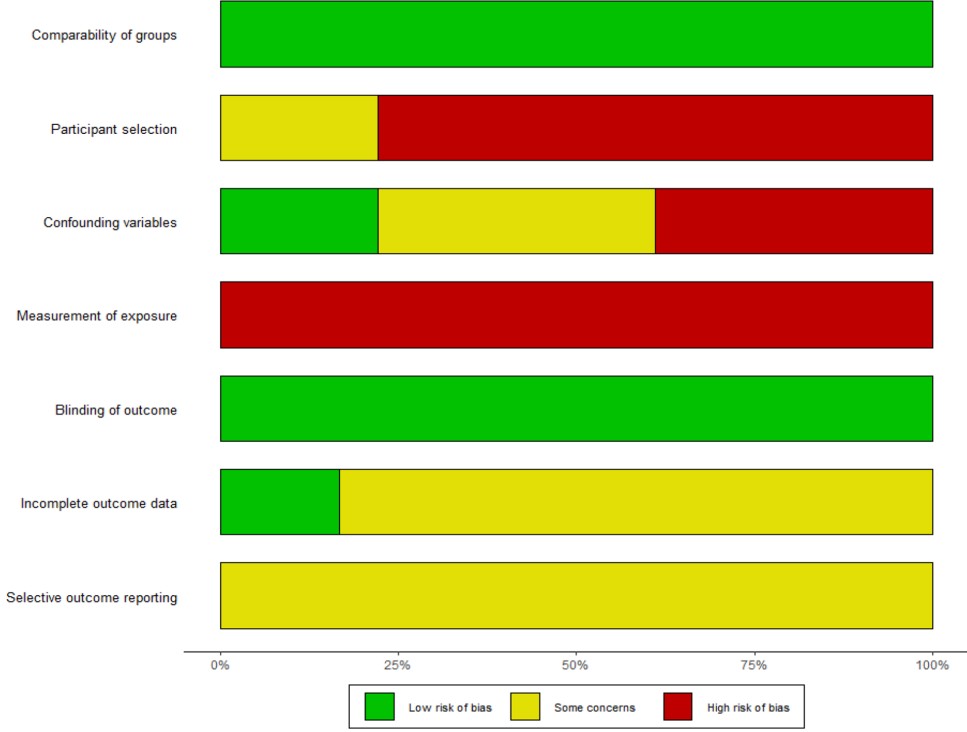

**Figure 2** Assessment of the risk of bias in identified studies using the RoBANS questionnaire. Author judgement for each RoBANS domain for the included articles. Green indicates a low risk of bias, yellow indicates unclear risk and red indicates a high risk of bias. The six domains are 'comparability of groups, participant selection, confounding variables, measurement of exposure, blinding of the outcome, incomplete outcome data, selective outcome reporting'. RoBANS, Risk of Bias Assessment Tool for Non-randomised Studies.

 

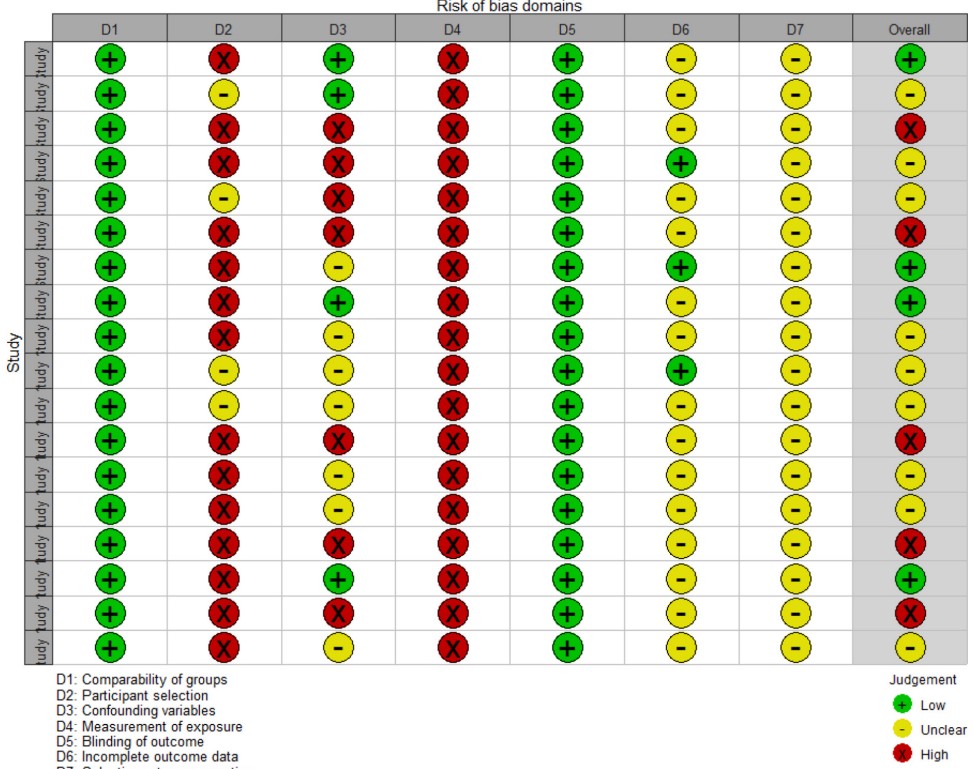

**Figure 3** Assessment summary of the risk of bias in identified studies using the RoBANS questionnaire. Author judgement for each RoBANS domain for the included articles. Green circles indicate low risk of bias, yellow circles indicate unclear risk and red circles indicate high risk of bias. The study numbers indicate the order of the articles presented in table 1. RoBANS, Risk of Bias Assessment Tool for Non-randomised Studies.

Eleven papers without full text and four papers not written in English were excluded. In addition, 413 papers were selected, excluding 102 papers that were not conducted in humans, 114 papers that were not conducted in adults and 110 duplicate papers after the first screening phase. Papers unrelated to tobacco (n=200) and COVID-19 (n=21) were excluded. In addition, a total of 18 papers were included, excluding 166 papers that were not related to tobacco behaviour and psychological states that changed during the COVID-19 pandemic as well as eight papers targeting COVID-19 patients or medical personnel in the first screening phase.

### Assessment result of the risk of bias

Thirteen papers were selected after excluding five papers with a high risk of bias, as determined through quality evaluation of the 18 papers using the RoBANS (figures 2 and 3).

### Characteristics of included papers by three subjects

All 13 selected papers were published in 2020; three were conducted in the USA, three in the UK, two in Italy, two in Turkey and one in China, India and Australia. All were cross-sectional design studies that used surveys. Comments, letters, editorials, viewpoints and correspondence were not included. In the included papers, the number of study participants varied from 93 to 52 002, and all participants were adults. We also presented the

average age of the participants in the papers or the ratio by age group. *Eight papers targeted cigarette-only users, and five reviewed e-cigarette and cigarette users together (table 1). Table 1 shows that smokers' smoking behaviour, cessation intentions and psychological states changed due to the COVID-19 pandemic, presented in all 13 papers. Seven papers studied changes in smoking behaviour, eight papers examined changes in smoking cessation and five papers examined changes in smokers' psychological state after the COVID-19 outbreak (table 1).*

### Integrating results of the 13 selected papers

Changes in frequency of smoking occurred in the early stages of the COVID-19 pandemic, regardless of cigarette type. However, smoking cessation intentions and attempts were higher among cigarette-only users than e-cigarette-only users. When the average age of participants was presented, and only papers surveyed according to the cigarette type were examined, the smoking behaviour of cigarette-only users was found to increase in the paper with an average age of 22. In a study targeting participants on average in their 50s or older, cigarette-only users showed a higher intention to quit smoking than e-cigarette-only users. Regardless of the average age of participants, in most studies, the amount of e-cigarette-only users' usage did not decrease, and the intention to quit smoking did not increase. In addition, it was found that all smokers, regardless of the tobacco product used,

**Table 1** Characteristics of included papers by three subjects

| First author, setting period | Age | Type of cigarette (N, %) | Subject 1* smoking behaviour | Subject 2* smoking cessation | Subject 3* psychological states |
|---|---|---|---|---|---|
| H.Tattan-Birch[13] England April–May | Mean (SD) 52.4 (17.6) | Total : 3179 -Cigarette-only users Never:1804 (56.7%) Long-term ex-cigarette users : 834 (26.2%) Recent ex-cigarette users : 72 (2.3%) Current smoker : 469 (14.8%) -E-cigarettes-only users no current use : 2987 (94%) current use : 192 (6%) | | Quit attempt ▲ Cigarette-only users: 12.2% ▲ E-cig: 10% | |
| E.K. Soule[29] US April | Mean (SD) 35.1 (10.8) | ▲ Total: E-cig use (n=93) 1. Dual user of E-cig and the following productsCigarettes: 50 2. Cigar: 11 3. Cigarillo or little cigar: 19 4. Smokeless: 7 5. Waterpipe: 11 | 1. Efforts to reduce the use of e-cigs have increased for dual users (Mean=3.71 vs 3.02; p<0.05), but e-cig use skills have increased (Mean=3.48 vs 2.27; p<0.05) and usage has increased (Mean=4.34 vs 3.72; p<0.05). 2. Users with high dependence on e-cigs have increased use of e-cigs compared with users with low dependence. (Mean=4.62 vs 3.35; p<0.001) | Users depended more on e-cigs had less effort to reduce their use of e-cigs (Mean=3.10, SD=0.63) than users who were less dependent on e-cigs (Mean=3.79, SD=0.61; p<0.02). | Users who rely heavily on e-cigs are more concerned about COVID-19 than users who do not rely heavily on e-cigs (Mean=3.86 vs 3.06; p<0.05). |
| Chertok[32] US Ohio April | Mean (SD) 33.5 (14.6) | Total: 810 1. Cigarettes-only: 115 (62.8%) 2. Dual users of cigarettes and e-cig: 68 (37.2%) 3. Not smoker: 627 | | (Not classified according to the type of cigarette but surveyed for all smokers.) ▲ Attempt to quit: 36.7% (n=66) ▲ Those who quit since COVID-19, those who desired to quit perceived a higher risk of infection. | |
| P.Caponnetto[14] Italy 2–26 April | Mean (SD) 34.7 (14.11) | Total: 1825 Dual user of cigarettes and E-cigarettes (64,3.5%) Cigarette and HTPs (33,1.8%), Cigarette-only users (582, 32%) HTPs-only user (81,4.4%) E-cig-only users (225, 12.3%) Former smoker (293, 16%) Never smoker (547, 30%) | 1. Dual users of cigarette and e-cig, and cigarette-only users' daily consumption has decreased. (Standardised residual (SR) value ▲ 2.7 and –1.9) 2.'Cigarette-only users' and the 'E-cig-only users' changed the way of purchasing products, and the majority of them changed their habits by having large stocks of cigarettes or e-liquids/cartridges at home avoid going out every day. | Most cigarette-only users (SR=11.6) have considered quitting E-cig-only users (SR=–4.2) have not considered stopping the use of e-cigs. | |

**Table 1** Continued

| First author, setting period | Age | Type of cigarette (N, %) | Subject 1* smoking behaviour | Subject 2* smoking cessation | Subject 3* psychological states |
|---|---|---|---|---|---|
| S.D. Kowitt[35] US 23 April–7 May | Mean (SD) 39.9 (13.4) | Total: 777 Cigarette-only users, 651 (83.8%) E-cig only 293 (37.3%) Smokeless tobacco only 170 (21.9%) Waterpipe tobacco only 103 (13.3%) | | | (Not classified according to the type of cigarette but surveyed for all smokers.) ▲ Having higher risk perceptions of COVID-19 was associated with higher quit intentions (B=0.38, p<0.001) ▲ Greater social distancing efforts were associated with higher quit intentions (B=0.11, p=0.01). -Higher odds of quitting attempts since COVID-19 started (aOR: 1.31, 95% CI: 1.04, 1.64). |
| R.Stanton[16] Australia April | Mean (SD) 50.5 (14.9) | Total=1491 Cigarettes-only=172 (11.5%) | | | Those who reported a negative change in smoking were more likely to have higher depression (aOR=1.09, 95% CI=1.04, 1.13), anxiety (aOR=1.12, 95% CI=1.06, 1.18), and stress (aOR=1.10, 95% CI=1.05, 1.15). |
| Jackson[36] England 21 March–20 April | 18–29: 19.5% 30's :15.0% 40's: 17.7% 50's: 17.5% 60's: 19.1% ≥70: 11.2% | Total: 53002 Ex-cigarette users Current cigarette users:13 602 (25.7%) Cigarette-only users: 8057 (15.2%) | | | Current and ex- cigarette users had higher odds than never smokers of reporting Significant stress about becoming seriously ill from the COVID-19 pandemic. Current cigarette users : OR=1.34, 95%CI=1.27–1.43; Ex- cigarette users : OR=1.22, 95%CI=1.16–1.28 |
| Jackson[37] England March–April | 18–24:13.4% 25–34:16.9% 35–44:15.7% 45–54:17.0% 55–64:14.5% ≥65: 22.5% | Cigarette-only users: 20558 | The COVID-19-19 lockdown was not associated with a significant change in smoking prevalence 17.0% (after) vs 15.9% (before), OR=1.09, 95% CI=0.95–1.24. | The COVID-19-19 lockdown was associated with increases in quit attempts 39.6 vs 29.1%, aOR=1.56, 95%CI=1.23–1.98, quit success 21.3 vs 13.9%, aOR=2.01, 95% CI=1.22–3.33 and cessation (8.8 vs 4.1%, aOR=2.63, 95% CI=1.69–4.09) among past-year cigarette-only users | |
| Kayhan Tetik[45] Turkey May | Mean (SD) 39.5(12.1) | Cigarette-only users: 357 | | 165 (46.2%) cigarette-only users of participants, 86 (24.1%) did not quit despite this pandemic. | |

Continued

**Table 1** Continued

| First author, setting period | Age | Type of cigarette (N, %) | Subject 1* smoking behaviour | Subject 2* smoking cessation | Subject 3* psychological states |
|---|---|---|---|---|---|
| Ren[30] China 14 February –29 March | Mean: 22.0 (min–max: 21.0–37.0) | Cigarette-only users:1172 | Some (30.1%) increased smoking during the pandemic. | | |
| Ozcelik[31] Turkey 1 January–30 June | Mean: 41.2 (min–max: 18–82) | Current cigarette-only users (N=111, 98%) Quitted smoking (N=3, 2%) | 77 people (67.5%) had no behavioural change associated with smoking | People with coronaphobia exhibited a significantly higher decrease or cessation of smoking than no change in smoking behaviour (OR: 18.22, 95% CI=2.30–144.31). | Decrease or cessation of smoking in people with coronaphobia was found to be significantly more significant in the individuals who had severe anxiety than in those who increased their smoking (OR:10.67, 95% CI=1.17–97.19). |
| R.Cancello[11] Italy 15 April– 4 May | 18–30:14.5% 31–60:65.1% ≥60: 20.4% | Total: 490 Cigarette-only users: 105 (21.4%) | Cigarette consumption increase: 38% | | |
| Chopra[12] India 15 August–30 August | Mean (SD) 33.33 (14.5) | No smoking: 939 (94.4%) Cigarette-only users: 56 | Smoking consumption is significantly reduced (Mean=0.02, p<0.05). | | |

*Subject 1 is a case in which a paper contains changes in smoking behaviour before and during the COVID-19 pandemic. Subject 2 is a case in which a paper contains perceptions of smoking or cessation intention before and during the COVID-19 pandemic. Subject 3 is a case in which a paper contains information on changes in psychological states, such as depression, stress and anxiety before and during the COVID-19 pandemic.
aOR, adjusted OR; HTPs, Heated Tobacco Products.

had increased stress and anxiety during the COVID-19 pandemic, and the more they did not intend to quit, the more stressed they were. We present detailed information on smoking behaviour, smoking cessation and psychological state changes according to cigarette type during the COVID-19 pandemic in the following six subheadings.

### Changes in smoking behaviour in e-cigarette users (or dual users) during the COVID-19 outbreak

According to Soule *et al*,[29] the higher the reliance on e-cigarettes, the higher the incidence of cigarette use. There have also been increasing efforts to reduce e-cigarette use among dual users. However, mastery of e-cigarette-specific behaviours during the additional time spent at home improved e-cigarette skills and increased e-cigarette use.[29] According to the article by Caponnetto *et al*,[14] daily cigarette consumption by dual users of cigarettes and e-cigarettes and cigarette users decreased slightly. Due to the COVID-19 outbreak, purchasing products, including tobacco products, has changed; individuals buy large numbers of products at once.

### Changes in smoking behaviour in cigarette-only users during the COVID-19 outbreak

According to the results of articles by Cancello *et al*[11] and Ren *et al*,[30] only cigarette consumption increased by 38% and 30%, respectively. Compared with before the COVID-19 pandemic, Chopra *et al* showed a significant decrease in smoking consumption (0.02 (0.03), p<0.05).[12] *However, in the study by* Ozcelik and Yilmaz Kara,[31] 67.5% of participants reported no change in smoking behaviour, which cannot be interpreted as a significant change in smoking behaviour during the COVID-19 pandemic.

### Changes in intentions and attempts to quit smoking among e-cigarette users (or dual users) during the COVID-19 outbreak

Tattan-Birch *et al*[13] found that a few cigarette-only users and e-cigarette-only users attempted to quit smoking due to the COVID-19 outbreak. Approximately, 1 in 10 current e-cigarette-only users reported an attempt to quit vaping because of COVID-19. Soule *et al* reported that users with high reliance on e-cigarettes exerted less effort to reduce product usage than users with low reliance.[29] According to a study by Caponnetto *et al*, while most cigarette-only users have considered quitting smoking, most e-cigarette-only users have not considered stopping e-cigarette use.[14] According to a study by Chertok *et al*, 36.7% of all smokers attempted to quit smoking during the pandemic regardless of cigarette type; people who quit smoking due to COVID-19 perceived smoking as a high-risk factor COVID-19 infection.[32]

### Changes in intentions and attempts to quit smoking among cigarette users during the COVID-19 outbreak

According to Jackson *et al*,[33] COVID-19 shutdowns have increased attempts to quit smoking. According to the results of the article by Kayhan Tetik *et al*,[34] which compared the success rates of smoking cessation before

and during the pandemic, the COVID-19 outbreak effectively promoted smoking cessation. Only 12.8% of people who quit smoking during the COVID-19 outbreak did not restart smoking. According to Ozcelik and Yilmaz Kara,[31] people with coronaphobia exhibited a significantly higher decrease or cessation in smoking than those with no change in smoking behaviour or increased smoking consumption. A study of cigarette-only users found that the number of attempts to quit smoking had increased in the USA.[32 35] In the UK, all studies conducted on cigarette-only users found that attempts to quit smoking had increased.[13 36] In the case of cigarette-only users, there are more successful quitting smoking cases than e-cigarette-only users.[13]

### The association between smoking and psychological changes in e-cigarette users (or dual users) during the COVID-19 outbreak

According to Soule *et al*, the greater the dependence on e-cigarettes, the higher the concern about COVID-19 infection than those with low dependence.[29] In the article by Kowitt *et al*, the higher the awareness regarding the risk of contracting COVID-19, regardless of cigarette type, the greater the willingness to quit (B=0.38, p<0.001) and the attempt to quit smoking (OR: 1.31, 95% CI 1.04 to 1.64).[35] In addition, as social distancing (B=0.11, p=0.01) was enforced more strongly, the willingness to quit smoking increased.[35]

### The association between smoking and psychological changes in cigarette users during the COVID-19 outbreak

Stanton *et al* found that people with increased tobacco consumption had a higher risk of depression, anxiety and stress.[16] Jackson *et al* reported that compared with non-smokers, current and former cigarette-only users had increased stress from COVID-19.[37] Ozcelik *et al* indicated that among people with coronaphobia, those who reduced or stopped smoking felt more anxious than those who increased their smoking frequency.[31]

## DISCUSSION

We examined the changes in smoking behaviour, cessation intentions and psychological states caused by the COVID-19 outbreak in smokers (cigarette-only users, e-cigarette-only users and dual users). Regardless of the cigarette type, the amount of smoking either increased or decreased in some cases. However, compared with e-cigarette-only users, most cigarette-only users' intentions and attempts to cease smoking increased. In addition, regardless of the type of cigarette, smokers showed a negative psychological state due to COVID-19. It was found that attempts and intentions to cease smoking increased as dependence on cigarettes decreased. Additionally, social distancing strengthened and awareness about COVID-19 increased.

The amount of smoking done by e-cigarette-only users increased with increased dependency. For dual users, those

who use cigarettes and e-cigarettes, efforts to reduce product usage increased; however, there was a tendency to buy cigarettes in bulk, increase e-cigarette usage skills and increase their use. This is thought to result from a tendency to recognise that using e-cigarettes is relatively less harmful to health than smoking cigarettes.[23] According to previous studies, many e-cigarette users recognise that e-cigarettes are relatively less harmful to health, their own and others,[23] because they contain far fewer harmful components[38] than cigarettes. However, it is not yet clear how e-cigarettes can help cigarette users quit smoking.[22] For this reason, e-cigarette users should consider the risks of using e-cigarettes as much as cigarettes.

Cigarette-only users' efforts to reduce product usage and quit smoking during the pandemic increased, especially when the average age group was high. In other words, the higher the age group, the higher the risk of COVID-19 infection and symptom severity. Therefore, it was found that the use of cigarettes decreased.[25] Efforts to reduce product usage by e-cigarette users, regardless of age, decreased. In particular, the higher the dependence of e-cigarette users, the higher were the worries related to COVID-19; however, efforts to reduce the amount of smoking were lower among users with low e-cigarette dependence. This could be because e-cigarette users think that using e-cigarettes is an attempt to quit smoking.[23] Therefore, e-cigarette users' intentions to quit smoking have not increased, even during the pandemic. A cohort analysis conducted in the USA found that e-cigarette users did not quit smoking, compared with cigarette-only users, and more than half of e-cigarette users still used e-cigarettes a year later.[39] In another study, it was unclear whether e-cigarette users were more likely to quit smoking.[40] Since many studies are still being conducted on the cessation effect of e-cigarettes, it should be recognised that e-cigarette use could be harmful to health.

Regardless of the cigarette type, the higher the awareness of the risks associated with contracting COVID-19, the higher the level of social distancing since the start of COVID-19 and intent to quit smoking. In the case of cigarette-only users, coronaphobia increased smoking cessation intentions. Among them, those who reduced smoking felt more anxious than those who increased their amount of smoking. Smokers were more likely to quit after understanding the health risks of smoking due to the 'vulnerability hypothesis', wherein smokers who care about their health would be inclined to quit smoking when they know it makes them vulnerable to certain diseases.[41 42] This presents the same context as the results of this systematic review.

In contrast, previous studies have shown that smokers have higher levels of depression, anxiety, stress and increased smoking amounts. According to a study by Klemperer et al,[43] 28.3% of patients reduced their tobacco use due to fear of infectious diseases, while 30.3% showed increased tobacco use and anxiety levels in both groups. However, in the literature surveyed in this study, it was found that the higher the level of anxiety, the higher the attempt to quit smoking. Therefore, further research exploring the relationship between mental health and smoking behaviours during the COVID-19 pandemic using standardised tools to measure mental health conditions, such as anxiety, is needed.

Increased awareness of COVID-19 risk has led to increased smoking cessation attempts, which can be considered a desirable outcome for physical health in the long run. However, for mental health, including anxiety, depression and increased stress during the pandemic, governments' response to address the increasing trend of smoking should be to conduct campaigns for risk awareness to protect their citizens' mental health. Research on the association between cigarette smoking and COVID-19 increases; the WHO also suggests that smokers may be considered at high risk of contracting COVID-19. However, because of the lack of research on the relationship between the use of liquid e-cigarettes and COVID-19, the willingness or attempts to quit e-cigarette use has not increased compared with cigarette-only users or dual users. Therefore, this study was able to review only cross-sectional studies. As smokers have a high risk of contracting certain diseases, both during and after the COVID-19 pandemic, further studies on the risk of e-cigarettes and their impact on smoking cessation, especially regarding COVID-19 infection, should be conducted.

This study is the first to systematically review the literature regarding changes in smoking behaviours, cessation intentions and psychological states of smokers, based on the study period, country and age of participants during the COVID-19 pandemic. Since this study aimed to investigate people's smoking behaviour, smoking cessation and psychological state in the early stages of the COVID-19 pandemic in 2020, all studies used questionnaire tools. There were no articles that had conducted qualitative or interventional studies on this topic. Therefore, only cross-sectional studies were included. This study only systematically reviewed articles published between January and December 2020. According to the results of this study, as social distancing and lockdown were strengthened in the early stages of the pandemic, smoking behaviour, smoking cessation intention and psychological changes appeared. However, in 2021, as vaccinations began to occur worldwide, social distancing decreased, as did people's willingness to participate in social distancing and stay-at-home.[44] Accordingly, smoking behaviour, smoking cessation and psychological status in 2021 may differ from that in 2020. When reviewing articles published in 2021, it is necessary to examine the changes in smoking behaviour, smoking cessation and psychological state of smokers while considering vaccine variables.

However, it is meaningful that this study looked at smoking usage behaviour during the COVID-19 pandemic based on cigarette type. This study could serve as primary data to establish effective strategies that focus on specific tobacco-product users by suggesting different smoking behaviours, smoking cessation intentions and psychological changes based on cigarette type.

## CONCLUSION

In this study, we examined changes in smoking behaviours, smoking cessation intentions and psychological states of smokers during COVID-19 according to cigarette type. We conducted a systematic review of 13 papers to suggest future research and policy directions that the government may implement for COVID-19 and smokers. During the COVID-19 pandemic, cigarette-only users' consumption did not change significantly, but dual users and e-cigarette-only users with a high dependence on e-cigarettes showed increased consumption. Cigarette users' quit intentions, attempts and cessation success rates increased, and even those with coronaphobia showed increased intent to quit smoking. On the other hand, users of e-cigarettes demonstrated fewer attempts to quit. As the COVID-19 pandemic continues, policies and campaigns should be implemented to increase the intent and attempt to stop smoking. Research and interventions on COVID-19, smoking and e-cigarette use should also be conducted in the medium to long term.

**Contributors** H-rC designed the study, analysed and interpreted the data, and wrote the manuscript. J-eH and EC contributed to the study design and interpretation of the results. H-rC and J-eH took full responsibility for the study and access to all data and controlled the decision to publish.

**Funding** This study was supported by the National Research Foundation of Korea (NRF) grant, funded by the Ministry of Science and ICT (MSIT) (Number 2020R1C1C1012562).

**Competing interests** None declared.

**Patient and public involvement** Patients and/or the public were not involved in the design, or conduct, or reporting, or dissemination plans of this research.

**Patient consent for publication** Not applicable.

**Ethics approval** Not applicable.

**Provenance and peer review** Not commissioned; externally peer reviewed.

**Data availability statement** All data relevant to the study are included in the article or uploaded as supplementary information.

**ORCID iDs**
Hae-ryoung Chun http://orcid.org/0000-0002-2567-2845
Ji-eun Hwang http://orcid.org/0000-0002-5094-6107

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
