## [Reviewer comments · BMJ Open]

ARTICLE DETAILS

TITLE (PROVISIONAL)	A systematic review of changed smoking behavior, smoking cessation, and psychological states of smokers according to cigarette type during the COVID-19 pandemic
AUTHORS	CHUN, HAE-RYOUNG; Cheon, Eunsil; Hwang, Ji-eun

VERSION 1 – REVIEW

REVIEWER	Koyama, Shihoko Osaka International Cancer Institute Cancer Control Center, Department of Cancer Epidemiology
REVIEW RETURNED	27-Aug-2021

GENERAL COMMENTS	The main research findings of this paper will be important for changed smoking behavior and psychological states of smoker during the COVID-19 pandemic. • I have found a few issues that, once addressed, will improve the manuscript. 1. Why did not use the term "heat not burn tobacco products" ? In some countries such as Korea, Portugal, Japan, Italy and Switzerland, some smoker changed from cigarettes to IQOS(heat not burn tobacco products). I am interested about smoking behavior changed in COVID-19 pandemic on deal user of cigarettes and HTPs.2. Page 3 line 22 "the daily use of e-cigarettes among e-cigarette smokers slightly increased" Please add reference.3. When did you search these term? In Prisma checklist No.6, "Specify the date when each source was last searched"
--

REVIEWER	Fujita, Yuko Kyushu Dental University
REVIEW RETURNED	24-Nov-2021

GENERAL COMMENTS	Although I think this review article is an important article, there are certain areas that require more attention. My comments are listed below: 1. The authors should describe the aim of this review and their hypothesis in Introduction section.2. The literature search and study selection should be specified in Method section.
---

	3. Did the authors search the references in each literature? 4. The authors should evaluate the publication bias of the studies. 5. How many reviewers assessed the risk bias of the studies? Additionally, authors should add the description how reviewers judged the risk bias of the studies in Method section.
--	---

REVIEWER	Garritsen, Heike Amsterdam UMC Locatie AMC, Department of Public and Occupational Health
REVIEW RETURNED	23-Dec-2021

GENERAL COMMENTS	Comments to authors General comments I suggest that the authors update their search as the current search was conducted in December 2020. It is very likely that in 2021 more studies on COVID-19 and smoking have been published. The entire manuscript would benefit from a grammar/spelling check. Title Comment 1. Type of cigarette/products used is missing in the title. Strengths and limitations Comment 2. Line 29, page 2. Please consider using “during” instead of “after”, as the COVID-19 pandemic is still going on. Abstract Comment 3. Data selection: “We included Articles on cigarettes...”. This could be more clear. You did not only include studies on cigarettes as a product, but on behavior as well. Introduction In general, the authors provide a lot of relevant information in the introduction. However, the introduction would benefit from a better structure in describing the research variables. It is quite unclear from the introduction what you actually want to investigate. Is it smoking behavior in general? Smoking cessation? Psychological pain? E-cigarettes? Type of cigarettes? Or all the above? The readability of the introduction would improve if sentences and paragraphs are more logically organized. Comment 4. Line 36-38, page 2. For being complete, it would be correct to mention that COVID-19 is an infectious disease. Comment 5. Line 45-55, page 2. This paragraph could be more clear. Overall, make sure to provide a complete and clear story when describing what is known about the relationship between smoking and COVID-19.  • Line 46: I believe that “smoking” does not have to be capitalized here. • Line 46: Please provide reference numbers after this sentence. • Line 51: Start your sentence with a capital letter or put “For example,” at the beginning of the sentence.
---

• Line 54-55: Give a time indication when talking about (non) changes of smoking rates. Do you mean since the start of the pandemic?

Comment 6. Line 57, page 2. You mention that a number of studies have been conducted regarding your research topic. What did these studies find? Also, references need to be added.

Comment 7. Line 6-8, page 2. "It is expected that smoking behavior will change given the difficulty in freely purchasing tobacco products as non-essential products, as they are locked down under the pandemic". However, in the Netherlands, cigarettes are sold in supermarkets, gas stations etc., and therefore still widely available. Is this different in other countries?

Comment 8. Line 18-20, page 3. The last sentence of this paragraph is somewhat confusing. What is meant by "psychological pain"? Does this refer to the stress, anxiety, and depression mentioned in the previous sentence? Please be more clear.

Comment 9. Line 18-20, page 3. What is meant by psychological pain AFTER COVID-19? Do you mean after being infected with COVID-19? Or during/after the COVID-19 pandemic?

Comment 10. Line 30, page 3. "Compared to studies examining..." Please consider starting a new paragraph with this sentence since the previous sentences are all about e-cigarettes and you're changing topic here.

Comment 11. Line 30. I am not sure whether "compared to" is the correct word choice in this sentence. You could say: "Although studies have examined whether..."

Comment 12. Line 34, page 3. You're talking about "psychological aspects" while earlier you were talking about "psychological pain" and in your title you are talking about "psychological states". Please be consistent in choice of words.

Methods

Comment 13. When was the review itself conducted?

Comment 14. Page 3, line 46-47. Please refer to the Supplementary file for PRISMA at the end of the sentence.

Comment 15. Line 54, page 3. Did you check references of included articles (snowballing)?

Comment 16. Line 54, page 3. Please refer to the Supplementary file in which the search strategies can be found.

Comment 17. Consider leaving 2.2 (patient and public involvement) out. Unless it is required by the journal.

Comment 18. Line 20, page 4. Paragraph "Inclusion and exclusion criteria". I think it's very well described why studies were excluded. However it's unclear what exactly were the inclusion criteria for studies to be included. Were there certain conditions that the studies had to meet to be included?

Comment 19. Line 24-25, page 4. "... that did not include humans or adults". This is the first time that you mention that your study is focusing on adults. Consider mentioning this at the end of your introduction as well.

Comment 20. Line 28-29, page 4. Please refer to the Supplementary file where the bias assessment can be found.

Comment 21. Line 38, page 4. Why is 'changes in psychological state' not mentioned as a main data point?

Comment 22. Line 45-46, page 4. "... which showed moderate reliability, promising feasibility, and validity." Does this apply to all included studies?

Results

The Results are hard to follow since they discuss each paper separately and in detail vs. integrating what was found in multiple papers and not comparing similarities and differences.

Comment 23. The results would benefit from a separate paragraph on differences between type of cigarette used.

Comment 24. Line 14, page 5. "...and psychology that changed after the pandemic." The term psychology is very broad. Do you mean the psychological pain/aspects? See also comment .. about being consistent in choice of words.

3.3. and 3.4. What does "by three participants" mean? This is unclear.

Comment 25. Line 28-29, page 5. I do not see why mentioning the months in which participants were studied is relevant. Consider leaving this out.

Comment 26. Line 32-33, page 5. "The only accepted papers were articles and not articles..." Please consider revising this sentence as it is unclear.

Comment 27. Line 35, page 5. "...and all participants were adults." This is obvious as being an adult was one of your inclusion criteria. Instead, you can mention the age range.

Comment 28. Line 45-46, page 5. "...eight papers that examined changes in perception of smoking." This was not among the variables you wanted to investigate, correct? Or what do you mean by 'perception of smoking'?

Comment 29. Line 56-57, page 5. What is meant by "e-cigarette skills are increasing" and why is this important?

Comment 30. Line 10-11, page 6. "The article by Chopra et al. also shows a significant decrease...". I think that using the word "also" is incorrect as the previous sentence is about an increase instead of decrease.

	Discussion Comment 31. It would be more logical to only mention your key findings in the first paragraph. The interpretation of your findings about e-cigarette use is now also discussed in this paragraph. I suggest to start a new paragraph for this. Comment 32. A strength & limitation section is missing. An example of a limitation is that you only included cross-section (survey) studies .
--	--

VERSION 1 – AUTHOR RESPONSE

Reviewers' Comments to the Authors

Reviewer: 1

Dr. Shihoko Koyama, Osaka International Cancer Institute Cancer Control Center

Comments to the Author:

The main research findings of this paper will be important for changed smoking behavior and psychological states of smoker during the COVID-19 pandemic.

- I have found a few issues that, once addressed, will improve the manuscript.

1. Why did not use the term "heat not burn tobacco products"?

In some countries such as Korea, Portugal, Japan, Italy and Switzerland, some smoker changed from cigarettes to IQOS(heat not burn tobacco products). I am interested about smoking behavior changed in COVID-19 pandemic on deal user of cigarettes and HTPs.

-> When we searched the articles, we used 'heat not burn' and 'heated tobacco product' keywords in the ScienceDirect database. When searching for articles using the MeSH term 'tobacco products,' literature search results did not change whether keywords related to HTP were added in PubMed. Therefore, when we performed a literature search in PubMed, we performed a comprehensive search for articles using the MeSH term 'tobacco products.'

-> We have added this sentence in the Literature Search Strategy section on page 8, lines 63-65. A literature search was conducted on PubMed using the following MeSH terms: "(depression OR stress OR anxiety OR psychology) AND ((tobacco OR tobacco products OR tobacco use OR electronic nicotine delivery system) AND COVID-19)" in the title or the abstract in 2020.

2. Page 3 line 22 "the daily use of e-cigarettes among e-cigarette smokers slightly increased" Please add reference.

-> We had added the reference at the beginning but have now moved the reference to the back of the sentence on page 6, lines 40–41.

The daily use of e-cigarettes among e-cigarette smokers slightly increased during the COVID-19 lockdown;¹⁷ therefore, it is necessary to investigate whether smoking behavior changes depending on cigarette type.

3. When did you search these term?

In Prisma checklist No.6, "Specify the date when each source was last searched"

-> We have included the systematic search dates in the Abstract.

[Our search was conducted in January 2021. We used the keywords COVID-19, cigarettes, electronic cigarettes (e-cigarettes), and psychological factors in PubMed and ScienceDirect and found papers published between January and December 2020.]

-> We have added the Search period in the Methods section on page 8, lines 58-61.

We searched articles using MeSH terms 'COVID-19, depression, stress, psychological, anxiety, tobacco, tobacco products, tobacco use, and electronic nicotine delivery system' in January 2021.

Only papers published between January and December 2020 were included in the study.

Reviewer: 2

Dr. Yuko Fujita, Kyushu Dental University

Comments to the Author:

Although I think this review article is an important article, there are certain areas that require more attention. My comments are listed below:

1. The authors should describe the aim of this review and their hypothesis in Introduction section.

-> We have added the aim and hypothesis of this review on page 7, lines 52-54, as in the sentence below.

Therefore, we conducted a systematic review based on the hypothesis that there would be changes in smoking behavior, cessation intention, and psychological states of adult smokers, depending on the tobacco product used during the COVID-19 pandemic.

2. The literature search and study selection should be specified in the Method section.

-> Literature search contents were included in the 'Literature Search Strategy' (part of the Methods section), and study selection contents were included in the 'Inclusion and Exclusion Criteria' and 'Data Extraction' sections of the Methods section.

3. Did the authors search the references in each literature?

-> We have added a description of the article search process on page 10, lines 97-101.

Using the backward snowballing method^{2, 3}, we checked the references of the articles that met the inclusion criteria. We checked the titles, journals, abstracts, and full texts of the references of the articles. We excluded articles that did not meet the inclusion criteria. In addition, articles not related to the subject of this study were excluded. As a result, no additional papers were added.

4. The authors should evaluate the publication bias of the studies.

-> The purpose of our study was to systematically review the literature to investigate research trends related to this research topic in the early stages of the COVID-19 pandemic. Therefore, publication bias was not considered because a meta-analysis was not performed.

5. How many reviewers assessed the risk bias of the studies? Additionally, authors should add the description how reviewers judged the risk bias of the studies in Method section.

-> Three reviewers assessed the risk of bias, and the method for judging bias is written in the Data Extraction section on page 9, lines 84-101.

Reviewer: 3

Miss Heike Garritsen, Amsterdam UMC Locatie AMC

Comments to the Author:

Comments to authors

General comments

I suggest that the authors update their search as the current search was conducted in December 2020. It is very likely that in 2021 more studies on COVID-19 and smoking have been published.

-> In January 2021, we searched for literature published between January and December 2020 and investigated changes in smoking behavior, smoking cessation, and psychological states in the early stages of the COVID-19 pandemic, the topic of this study. As vaccines rolled out globally in 2021, new measures and policies to respond to COVID-19 have emerged. If we considered these variables in this review, we believe that our topic would be blurred.

-> A review of the literature published after 2021 is planned as a follow-up study.

-> We have written that reviewing only the literature published in 2020 as a limitation on page 23, lines 285-291.

Second, this study systematically reviewed only articles published between January and December 2020. As the vaccine was introduced in 2021, different policies to respond to COVID-19 were implemented in 2020. As a result, smoking behavior, smoking cessation, and psychological state in 2021 will differ to 2020. When reviewing articles published in 2021, it is necessary to examine the changes in smoking behavior, smoking cessation, and psychological state of smokers while considering vaccine variables.

The entire manuscript would benefit from a grammar/spelling check.

-> On April 2, 2021, this manuscript was edited for language and grammar accuracy by Editage, which specializes in editing documents for international journals published in English. We have attached the editing certificate to the last page of this Word file.

-> We have ensured that the grammar, spelling, and flow corrections of the entire manuscript have been reviewed and corrected, including sentences that were modified according to the reviewer's comments.

Title

Comment 1. Type of cigarette/products used is missing in the title.

-> We investigated changed smoking behaviors, cessation intentions, and psychological states of smokers according to 'type of cigarette'. Therefore, we have changed the title to: A systematic review of changed smoking behavior, smoking cessation, and psychological states of smokers according to cigarette type during the COVID-19 pandemic.

Strengths and limitations

Comment 2. Line 29, page 2. Please consider using "during" instead of "after", as the COVID-19

pandemic continues.

-> We have revised it to 'during the COVID-19 pandemic' instead of 'after the COVID-19 pandemic' on pages 3-4.

Abstract

Comment 3. Data selection: "We included Articles on cigarettes...". This could be more clear. You did not only include studies on cigarettes as a product, but on behavior as well.

-> We have revised it to: *We included articles in full text, written in English, and that surveyed adults. The topics included smoking behavior, smoking cessation, psychological state of smokers, and COVID-19-related topics.*

Introduction

In general, the authors provide a lot of relevant information in the introduction. However, the introduction would benefit from a better structure in describing the research variables. It is quite unclear from the introduction what you actually want to investigate. Is it smoking behavior in general? Smoking cessation? Psychological pain? E-cigarettes? Type of cigarettes? Or all the above? The readability of the introduction would improve if sentences and paragraphs are more logically organized.

-> The purpose of this study was to investigate the smoking behavior, cessation intention, and psychological state of smokers according to the types of cigarettes used in the early stages of the COVID-19 pandemic. The Introduction section has been reorganized as follows:

Paragraph 1: COVID-19 pandemic

Paragraph 2: Smokers are at high risk for COVID-19 infection.

Paragraph 3: There is insufficient research on the smoking behavior and cessation intention of smokers during the COVID-19 pandemic.

Paragraph 4: There is not enough research to examine the psychological state of smokers during the COVID-19 pandemic.

Paragraph 5: There is no systematic review on smokers' smoking behavior, cessation intention, and psychological state according to cigarette type during the COVID-19 pandemic.

Paragraph 6: Aim and expectations of this study

Comment 4. Line 36-38, page 2. For being complete, it would be correct to mention that COVID-19 is an infectious disease.

-> We have changed 'a disease ~' to 'an infectious disease. on page 5, line 3.

Comment 5. Line 45-55, page 2. This paragraph could be more clear. Overall, make sure to provide a complete and clear story when describing what is known about the relationship between smoking and COVID-19.

- Line 46: I believe that "smoking" does not have to be capitalized here.

-> We have changed it to lowercase on page 5, line 10.

- Linde 46: Please provide reference numbers after this sentence.

-> On page 5, lines 9-10, we have added a reference and modified the sentence as follows:

Smoking is a high-risk factor for COVID-19 infection and there are many systematic reviews and meta-analyses that report the association between COVID-19 and smoking.⁴⁻⁶

- Line 51: Start your sentence with a capital letter or put "For example," at the beginning of the sentence.

-> We have changed it to a capital letter.

- Line 54-55: Give a time indication when talking about (non) changes of smoking rates. Do you mean since the start of the pandemic?

-> On page 5, lines 18-19, we have changed the sentence to 'A few investigations have been conducted on whether smoking behavior⁸ and cessation efforts⁹ have changed during the COVID-19 pandemic.'

Comment 6. Line 57, page 2. You mention that a number of studies have been conducted regarding your research topic. What did these studies find? Also, references need to be added.

-> On page 5, lines 18-21, we have added a reference and modified the sentence as follows:

A few investigations have been conducted on whether smoking behavior⁷ and cessation efforts⁸ have changed during the COVID-19 pandemic. Previous studies have shown that smoking has decreased⁷, and the number of quitting attempts has increased⁸ since the COVID-19 pandemic.

Comment 7. Line 6-8, page 2. "It is expected that smoking behavior will change given the difficulty in freely purchasing tobacco products as non-essential products, as they are locked down under the pandemic". However, in the Netherlands, cigarettes are sold in supermarkets, gas stations etc., and therefore still widely available. Is this different in other countries?

-> We found that articles reported that people were buying in bulk if the lockdown made it difficult for them to buy cigarettes. We have marked these references. Considering that not all countries are like this, on page 6, lines 23-26, we have modified the sentence as follows:

In some countries, it is expected that smoking behavior will change given the difficulty in freely purchasing tobacco products as non-essential items; individuals are locked down under the pandemic and are required to stay at home for prolonged periods.⁹

Comment 8. Line 18-20, page 3. The last sentence of this paragraph is somewhat confusing. What is meant by "psychological pain"? Does this refer to the stress, anxiety, and depression mentioned in the previous sentence? Please be more clear.

-> We referred to 'psychological pain' as stress, anxiety, and depression in the previous sentence. We have changed 'psychological pain' to 'negative psychological states' for clarity on pages 6, lines 30-32, and 35-37.

There are currently insufficient studies on smokers' negative psychological states during the COVID-19 pandemic. Further research is needed because the relationship between negative psychological states and health behavior is unclear.¹⁰

Previous research has shown that negative psychological states, such as stress, anxiety, and depression can influence tobacco use behavior.¹¹

Comment 9. Line 18-20, page 3. What is meant by psychological pain AFTER COVID-19? Do you mean after being infected with COVID-19? Or during/after the COVID-19 pandemic?

-> We defined 'psychological states' as stress, anxiety, and depression caused by lockdown and social distancing during the COVID-19 pandemic.

Comment 10. Line 30, page 3. "Compared to studies examining..." Please consider starting a new paragraph with this sentence since the previous sentences are all about e-cigarettes and you're changing topic here.

-> We have started the sentence in a new paragraph on page 7, line 47.

Many studies have investigated whether smoking is a high-risk factor for COVID-19.^{5, 6} However, relatively few studies have been published on whether smoking behavior, cessation intention, and psychological state have changed during the COVID-19 pandemic. *In the future, it is expected that this study will establish a scientific basis and policy response to smoking behavior, cessation intention, and psychological state based on cigarette type, both during and post COVID-19.*

Therefore, we conducted a systematic review based on the hypothesis that there would be changes in smoking behavior, cessation intention, and psychological states of adult smokers, depending on the tobacco product used during the COVID-19 pandemic.

Comment 11. Line 30. I am not sure whether "compared to" is the correct word choice in this sentence. You could say: "Although studies have examined whether..."

-> Although many studies are investigating whether smoking is a high-risk factor for COVID-19, we wanted to write about the lack of studies on whether changes in smoking behavior, smoking cessation intention, and psychological states occurred during the COVID-19 pandemic. Therefore, on page 7, lines 47-49, we have modified the sentence as follows:

Many studies have investigated whether smoking is a high-risk factor for COVID-19. However, relatively few studies have been published on whether smoking behavior, cessation intention, and psychological state have changed during the COVID-19 pandemic.

Comment 12. Line 34, page 3. You're talking about "psychological aspects" while earlier you were talking about "psychological pain" and in your title you are talking about "psychological states". Please

be consistent in choice of words.

-> We have modified all terms to 'psychological state' to be consistent.

Methods

Comment 13. When was the review itself conducted?

-> On page 8, lines 58-61, we have presented the following sentence:

We searched articles using MeSH terms 'COVID-19, depression, stress, psychological, anxiety, tobacco, tobacco products, tobacco use, and electronic nicotine delivery system' in January 2021.

Only papers published between January and December 2020 were included in the study.

Comment 14. Page 3, line 46-47. Please refer to the Supplementary file for PRISMA at the end of the sentence.

-> On page 8, lines 57-58, we have changed the sentence to: We systematically reviewed the data according to the Preferred Reporting Items for Systematic Reviews and Meta-Analyses (PRISMA), shown in the Supplementary file.

Comment 15. Line 54, page 3. Did you check references of included articles (snowballing)?

-> On page 10, lines 97-101, we have added a description of the article search process

Using the backward snowballing method,^{2, 3} we checked the references of the articles that met the inclusion criteria. We checked the titles, journals, abstracts, and full texts of the references of the

articles. We excluded articles that did not meet the inclusion criteria. In addition, articles not related to the subject of this study were excluded. As a result, no additional papers were added.

Comment 16. Line 54, page 3. Please refer to the Supplementary file in which the search strategies can be found.

-> On page 8, line 61-62, we have changed the sentence to: All search terms are listed in Supplementary Table 1.

Comment 17. Consider leaving 2.2 (patient and public involvement) out. Unless it is required by the journal.

-> We have deleted this information.

Comment 18. Line 20, page 4. Paragraph "Inclusion and exclusion criteria". I think it's very well described why studies were excluded. However it's unclear what exactly were the inclusion criteria for studies to be included. Were there certain conditions that the studies had to meet to be included?

-> We included all articles that did not meet the exclusion criteria. In other words, the inclusion criteria was that the articles should be written in English, include humans, and be related to smoking

behavior, cessation intention, and a negative psychological state during COVID-19. Therefore, we have added the following sentence on page 8, line 73:

The inclusion criteria were cases that did not meet the exclusion criteria.

Comment 19. Line 24-25, page 4. "... that did not include humans or adults". This is the first time that you mention that your study is focusing on adults. Consider mentioning this at the end of your introduction as well.

-> In the research purpose section of the Introduction, we added that only articles that included adults were reviewed on page 7, lines 52-54.

Therefore, we conducted a systematic review based on the hypothesis that there would be changes in smoking behavior, cessation intention, and psychological states of adult smokers, depending on the tobacco products used during the COVID-19 pandemic.

Comment 20. Line 28-29, page 4. Please refer to the Supplementary file where the bias assessment can be found.

-> We specified that the risk of bias assessment results can be seen in Figures 2 and 3. On page 9, lines 79-81, the sentence has been revised as follows: Finally, articles with a high risk of bias were excluded using the Risk of Bias Assessment Tool for Non-randomized Studies (RoBANS) (Figures 2 and 3).¹²

Comment 21. Line 38, page 4. Why is 'changes in psychological state' not mentioned as a main data point?

-> We have added 'changes in psychological state. On page 9, lines 86-89, we have revised the sentence to: The main data points included were the following: study details (author, journal, publication date, country, study design, study period, and funding), total number of participants, types of cigarettes, changes in smoking behavior, cessation intention, and psychological state.

Comment 22. Line 45-46, page 4. "... which showed moderate reliability, promising feasibility, and validity." Does this apply to all included studies?

-> We cited published articles that described the RoBANS tool as having moderate reliability, promising feasibility, and validity. We have deleted this description.

On page 9, lines 90-91, we have changed the sentence to: Two authors (HR and ES) independently assessed the quality of included studies using the RoBANS ~~which showed moderate reliability, promising feasibility, and validity.~~

Results

The Results are hard to follow since they discuss each paper separately and in detail vs. integrating what was found in multiple papers and not comparing similarities and differences.

-> We have divided and presented the integrated results of the 13 selected papers under six subheadings, providing details of individual papers separately, to make it easy to compare the contents. On pages 11-12, lines 132–143, we present the integrated results as follows:

3.4. Integrating results of the 13 selected papers

In the early stages of the COVID-19 pandemic, behavioral changes did not appear to be the same, such as increasing or decreasing the amount of cigarettes smoked, regardless of cigarette type. However, smoking cessation intentions and attempts were higher among cigarette users than e-cigarette users. In addition, if the average age of the study participants was ≥ 40 years, the number of attempts to quit smoking was higher. In the case of those in their 30s, the higher the awareness of COVID-19 infection, the higher the attempts to quit smoking. In addition, it was found that all smokers, regardless of the tobacco product used, had increased stress and anxiety during the COVID-19 pandemic, and the more they did not intend to quit, the more stressed they were. We present detailed information on smoking behavior, smoking cessation, and psychological state changes according to cigarette type during the COVID-19 pandemic in the six sub-headings that follow.

Comment 23. The results would benefit from a separate paragraph on differences between type of cigarette used.

-> We have revised the Results section and present the differences according to cigarette type in separate paragraphs. We divided and presented the differences in smoking behavior, smoking cessation, and psychological state according to cigarette type under six sub-headings.

Comment 24. Line 14, page 5. "...and psychology that changed after the pandemic." The term psychology is very broad. Do you mean the psychological pain/aspects? See also comment .. about being consistent in choice of words.

-> On page 10, lines 109-113, we have revised the term "psychology" to "psychological states". We have revised the sentence to: *Papers unrelated to tobacco (n = 200) and COVID-19 (n = 21) were excluded. In addition, a total of 18 papers were included, excluding 166 papers that were not related to tobacco behavior and psychological states that changed during the COVID-19 pandemic, as well as eight papers targeting COVID-19 patients or medical personnel in the first screening phase.*

3.3. and 3.4. What does "by three participants" mean? This is unclear.

-> We have corrected the typographical errors. We changed the "by three participants" to "by three subjects."

Comment 25. Line 28-29, page 5. I do not see why mentioning the months in which participants were studies is relevant. Consider leaving this out.

-> On page 11, lines 120-121, we have deleted this content. We changed the sentence to: *All 13 selected papers were published in 2020; three papers were conducted in the US, three in the UK, two in Italy, two in Turkey, and one each in China, India, and Australia.*

Comment 26. Line 32-33, page 5. "The only accepted papers were articles and not articles..." Please consider revising this sentence as it is unclear.

-> On page 11, lines 122-123, we have revised the sentence to: *Comments, letters, editorials, viewpoints, and correspondence were not included.*

Comment 27. Line 35, page 5. "...and all participants were adults." This is obvious as being an adult was one of your inclusion criteria. Instead, you can mention the age range.

-> Table 1 shows the average age of the participants presented in the study or the ratio by age group. The following sentence has been added on page 11, lines 124-125.

We also presented the average age of the participants in the papers or the ratio by age group.

Comment 28. Line 45-46, page 5. "...eight papers that examined changes in perception of smoking." This was not among the variables you wanted to investigate, correct? Or what do you mean by 'perception of smoking'?

-> The purpose of this study was to investigate changes in smoking behavior, smoking cessation, and psychological states. We have corrected the typographical errors. On page 11, lines 128-130, we changed the term "perception of smoking" to "*smoking cessation.*"

Seven papers studied changes in smoking behavior, eight papers examined changes in smoking cessation, and five papers examined changes in smokers' psychological state after the COVID-19 outbreak (Table 1).

Comment 29. Line 56-57, page 5. What is meant by "e-cigarette skills are increasing" and why is this important?

-> Citing the contents of the reference, we have revised the sentence as follows on page 12, lines 149-150:

However, mastery of e-cigarette-specific behaviors during the additional time spent at home improved e-cigarette skills and increased e-cigarette use.¹³

Comment 30. Line 10-11, page 6. "The article by Chopra et al. also shows a significant decrease...". I think that using the word "also" is incorrect as the previous sentence is about an increase instead of decrease.

-> In accordance with the reviewer's comment, we have deleted the word 'also' on page 13, lines 158-159.

[Compared to before the COVID-19 pandemic Chopra et al. ~~also showed a significant decrease in the amount of smoking consumption (0.02 [0.03], $P < 0.05$).~~ ^{7]}

Discussion

Comment 31. It would be more logical to only mention your key findings in the first paragraph. The interpretation of your findings about e-cigarette use is now also discussed in this paragraph. I suggest to start a new paragraph for this.

-> We have only mentioned key findings in the first paragraph on page 20, lines 218-225.

We examined the changes in smoking behavior, cessation intentions, and psychological states caused by the COVID-19 outbreak in smokers (cigarette users, e-cigarette users, and dual users). Regardless of the cigarette type, in some cases, the amount of smoking either increased or decreased. However, compared to e-cigarette users, tobacco users' intentions and attempts to cease smoking increased. In addition, regardless of the type of cigarette, smokers showed a negative psychological state due to COVID-19. It was found that attempts and intentions to cease smoking increased as dependence on cigarettes decreased. Additionally, social distancing strengthened and awareness about COVID-19 increased.

-> In accordance with the reviewer's comment, we have changed the interpretation of findings about e-cigarette use and started a new paragraph on page 20, lines 226-235.

The amount of smoking done by e-cigarette-only users increased with increased dependency. For dual users, those who use cigarettes and e-cigarettes, efforts to reduce product usage increased; however, there was a tendency to buy cigarettes in bulk, increase e-cigarette usage skills, and increase their use. This is thought to be the result of a tendency to recognize that using e-cigarettes is relatively less harmful to health than smoking cigarettes.¹⁴ According to previous studies, many e-cigarette smokers recognize that e-cigarettes are relatively less harmful to health, their own and others',¹⁴ because they contain far fewer harmful components¹⁵ than cigarettes. However, it is not yet clear to what extent e-cigarettes can help smokers quit smoking.¹⁶ For this reason, e-cigarette users should consider the risks of using e-cigarettes as much as cigarettes.

Comment 32. A strength & limitation section is missing. An example of a limitation is that you only included cross-section (survey) studies .

-> On page 23, lines 281-292, we have added the strength and limitation section.

->We have added that only articles published between January and December 2020 were reviewed, as our topic covered only the initial situation during the COVID-19 pandemic.

5. STRENGTH AND LIMITATION

A limitation of this study is that only cross-sectional studies were included. Since the purpose of this study was to investigate people's smoking behavior, smoking cessation, and psychological state in the early stages of the COVID-19 pandemic in 2020, all studies used questionnaire tools. There were no articles that had conducted qualitative or interventional studies on this topic at the time. Second, this study systematically reviewed only articles published between January and December 2020. As the vaccine was introduced in 2021, different policies to respond to COVID-19 were implemented in 2020. As a result, smoking behavior, smoking cessation, and psychological state in 2021 will differ to 2020. When reviewing articles published in 2021, it is necessary to examine the changes in smoking behavior, smoking cessation, and psychological state of smokers while considering vaccine variables. However, it is meaningful that this study looked at smoking usage behavior during the COVID-19 pandemic based on cigarette type.

Reviewer: 1

Competing interests of Reviewer: No

Reviewer: 2

Competing interests of Reviewer: I have no competing interests.

Reviewer: 3

Competing interests of Reviewer: None

Editor(s)' Comments to Author (if any):

1. Adriaens K, Van Gucht D, Baeyens F. Differences between dual users and switchers center around vaping behavior and its experiences rather than beliefs and attitudes. *International journal of environmental research and public health*. 2018;15(1):12.
2. Wohlin C. Guidelines for snowballing in systematic literature studies and a replication in software engineering. 2014:1-10.
3. Greenhalgh T, Peacock R. Effectiveness and efficiency of search methods in systematic reviews of complex evidence: audit of primary sources. *Bmj*. 2005;331(7524):1064-1065.
4. Ni Y, Shi G, Qu J. Indoor PM_{2.5}, tobacco smoking and chronic lung diseases: A narrative review. *Environmental research*. 2020;181:108910.
5. Vardavas CI, Nikitara K. COVID-19 and smoking: A systematic review of the evidence. *Tobacco induced diseases*. 2020;18
6. Umnuaypornlert A, Kanchanasurakit S, Lucero-Prisno DEI, Saokaew S. Smoking and risk of negative outcomes among COVID-19 patients: A systematic review and meta-analysis. *Tobacco induced diseases*. 2021;19
7. Chopra S, Ranjan P, Singh V, et al. Impact of COVID-19 on lifestyle-related behaviours-a cross-sectional audit of responses from nine hundred and ninety-five participants from India. *Diabetes & Metabolic Syndrome: Clinical Research & Reviews*. 2020;14(6):2021-2030.
8. Tattan-Birch H, Perski O, Jackson S, Shahab L, West R, Brown J. COVID-19, smoking, vaping and quitting: a representative population survey in England. *Addiction*. 2020;
9. Caponnetto P, Inguscio L, Saitta C, Maglia M, Benfatto F, Polosa R. Smoking behavior and psychological dynamics during COVID-19 social distancing and stay-at-home policies: A survey. *Health psychology research*. 2020;8(1)
10. Stanton R, To QG, Khalesi S, et al. Depression, Anxiety and Stress during COVID-19: Associations with Changes in Physical Activity, Sleep, Tobacco and Alcohol Use in Australian Adults. *Int J Environ Res Public Health*. Jun 7 2020;17(11)doi:10.3390/ijerph17114065
11. García-Álvarez L, Fuente-Tomás L, Sáiz PA, García-Portilla MP, Bobes J. Will changes in alcohol and tobacco use be seen during the COVID-19 lockdown? *Adicciones*. Apr 1 2020;32(2):85-89. ¿Se observarán cambios en el consumo de alcohol y tabaco durante el confinamiento por COVID-19? doi:10.20882/adicciones.1546
12. Kim SY, Park JE, Lee YJ, et al. Testing a tool for assessing the risk of bias for nonrandomized studies showed moderate reliability and promising validity. *Journal of clinical epidemiology*. 2013;66(4):408-414.
13. Soule EK, Mayne S, Snipes W, Guy MC, Breland A, Fagan P. Impacts of COVID-19 on electronic cigarette purchasing, use and related behaviors. *International Journal of Environmental Research and Public Health*. 2020;17(18):6762.
14. Etter JF, Bullen C. Electronic cigarette: users profile, utilization, satisfaction and perceived efficacy. *Addiction*. 2011;106(11):2017-2028.
15. Goniewicz ML, Knysak J, Gawron M, et al. Levels of selected carcinogens and toxicants in vapour from electronic cigarettes. *Tobacco control*. 2014;23(2):133-139.
16. Grana RA, Popova L, Ling PM. A longitudinal analysis of electronic cigarette use and smoking

cessation. *JAMA internal medicine*. 2014;174(5):812-813.

VERSION 2 – REVIEW

REVIEWER	Koyama, Shihoko Osaka International Cancer Institute Cancer Control Center, Department of Cancer Epidemiology
REVIEW RETURNED	04-Feb-2022

GENERAL COMMENTS	Thank you for opportunity. These studies are important for understand the relationship between smoking behavior and COVID-19. I hope that my comment is very useful for the improvement of the article. Major  • I know cigarette smoking is a high-risk factor for COVID-19. However, I'm sorry for not knowing it is same in e-cig smoking. Did your citation studies include e-cig smoking result? • I think we didn't know the risk for COVID-19 in April 2020. Your citation study (Ref No.5) was published in 2021. Therefore, I felt the second part of introduction has a contradiction. I want you to give a careful introduction and consideration in chronological order. • I think older people are using cigarettes rather than e-cigarettes. Older people may have quit smoking because they are at higher risk of COVID-19. So, please discuss smoking and the effects of age. Minor  • For the whole study, to use term "smoker" is unclear. Please revise "smoker" to "cigarette user" or "e-cig user" or "dual user". • For the same reason as above, "tobacco consumption" is unclear. Please show what type of tobacco consumption. Intro  • Line 6 Please correct citation for website. Results 3.4.5. and 3.4.6. were same title. Is it correct? Table  • Table 1 is hard to understand. Please improve the font and size as a whole to make it easier to understand. • E.K.Soule 's study; What means "M" in subject 1 column? • P.Caponnetto's study; The way of writing (number and percentage) do not similarly other studies. If you want to express the same thing, please use the same expression and unify. • Is "cigarette only" and "only cigarette users" same?
---

	 • The P-value and 95% confidence interval are written differently for each paper, making it difficult to read. Strength and limitation  • line286 It was written about vaccines. Why important the relationship of smoking and vaccination? • There is almost no mention of the strengths of this research, but what are the strengths? Others  • Why did attach the yellow highlighted PDF file? What does it mean?
--	---

REVIEWER	Fujita, Yuko Kyushu Dental University
REVIEW RETURNED	31-Jan-2022

GENERAL COMMENTS	The comments were satisfactorily addressed, and I anticipate acceptance of the revised manuscript.
--

REVIEWER	Garritsen, Heike Amsterdam UMC Locatie AMC, Department of Public and Occupational Health
REVIEW RETURNED	08-Feb-2022

GENERAL COMMENTS	I thank the authors for taking into considerations my comments and altering the manuscript.
---

VERSION 2 – AUTHOR RESPONSE

Reviewer: 2

Dr. Yuko Fujita, Kyushu Dental University

Comments to the Author:

The comments were satisfactorily addressed, and I anticipate acceptance of the revised manuscript.

-> Thank you for your great comments and answer.

Reviewer: 1

Dr. Shihoko Koyama, Osaka International Cancer Institute Cancer Control Center

Comments to the Author:

Thank you for opportunity.

These studies are important for understand the relationship between smoking behavior and COVID-19.

I hope that my comment is very useful for the improvement of the article.

Major

- I know cigarette smoking is a high-risk factor for COVID-19. However, I'm sorry for not knowing it is same in e-cig smoking. Did your citation studies include e-cig smoking result?

-> The papers we cited in the second paragraph of the introduction do not include the result that e-cigarette use is a risk factor for COVID-19. Therefore, we have cited the following papers suggesting that although there is no literature that cigarettes are the cause of COVID-19 infections, e-cigarette use is vulnerable to virus infection and could cause severe COVID-19 symptoms on page 4, lines 15-18.

□ Similar to cigarette use, e-cigarette use can cause oxidative stress and inflammatory response in the lungs, making e-cigarette users more susceptible to bacterial or viral infections. 1 Although no literature indicates that cigarettes are the direct cause of COVID-19 infection, smokers have a higher risk of severe COVID-19 symptoms than nonsmokers. 2

- I think we didn't know the risk for COVID-19 in April 2020. Your citation study (Ref No.5) was published in 2021. Therefore, I felt the second part of introduction has a contradiction. I want you to give a careful introduction and consideration in chronological order.

->The Citation study* is a systematic review and meta-analysis paper that reviewed the original articles about the high risk of COVID-19 of smoking published in 2020. The Original articles analyzed the data of the early period of the COVID-19 Pandemic between January and April 2020.

*Umnuaypornlert, A., Kanchanasurakit, S., Lucero-Prisno, D. E. I., & Saokaew, S. (2021). Smoking and risk of negative outcomes among COVID-19 patients: A systematic review and meta-analysis. Tobacco induced diseases, 19.

->For your information, the study's reference number was changed because we additionally cited two new studies in the first paragraph of the introduction.

- I think older people are using cigarettes rather than e-cigarettes. Older people may have quit smoking because they are at higher risk of COVID-19. So, please discuss smoking and the effects of age.

-> As requested, we added the following statement on page 6, lines 49 - 54, citing the previous paper.

->In addition, after the COVID-19 pandemic, smoking behavior may change by age depending on the cigarette type. The use of e-cigarettes increases among persons in their early 20s compared with old persons.³ In addition, it is known that older persons are more vulnerable to COVID-19 infection and can develop severe infection symptoms. ⁴ Therefore, there may be an increase in the older persons' intention to quit smoking cigarettes. However, young people who use e-cigarettes may not be less willing to quit smoking.

-> We presented the changes in smoking behavior and smoking cessation by average age since the COVID-19 pandemic in “3.4. Integrating results of the 13 selected papers” on pages 10-11, lines 145-149.

-> When the average age of participants was presented, and only papers surveyed according to the cigarette type were examined, the smoking behavior of cigarette-only users was found to increase in the paper with an average age of 22. In a study targeting participants on average in their 50s or older, cigarette-only users showed a higher intention to quit smoking than e-cigarette-only users. Regardless of the average age of participants, in most studies, the amount of e-cigarette-only users' usage did not decrease, and the intention to quit smoking did not increase.

-> As requested, we have added these sentences in the Discussion on pages 19-20, lines 245-249.

-> Cigarette-only users' efforts to reduce product usage and quit smoking during the pandemic increased, especially when the average age group was high. In other words, the higher the age group, the higher the risk of COVID-19 infection and symptom severity. Therefore, it was found that the use of cigarettes decreased. 4 Efforts to reduce product usage by e-cigarette users, regardless of age, decreased.

Miner

- For the whole study, to use term “smoker” is unclear. Please revise “smoker” to “cigarette user” or “e-cig user” or “dual user”.

-> As requested, the term smoker was used to refer to all people who smoke regardless of type collectively, and when using a specific cigarette, it was divided into "cigarette user," "e-cig user," or "dual user."

- For the same reason as above, “tobacco consumption” is unclear. Please show what type of tobacco consumption.

-> As requested, we have changed to ‘only cigarette consumption’ instead of ‘tobacco consumption.’

Intro

- Line 6 Please correct citation for website.

-> As requested, we have corrected the citation.

Results

3.4.5. and 3.4.6. were same title. Is it correct?

->We have changed the 3.4.6. title: 3.4.6. The association between smoking and psychological changes in cigarette users during the COVID-19 outbreak

Table

- Table 1 is hard to understand. Please improve the font and size as a whole to make it easier to understand.

-> As requested, we have improved table 1's font and size to make it easier to understand.

- E.K.Soule 's study; What means "M" in subject 1 column?

-> "M" means Mean rating. We have changed it to "mean" instead of "m" to make it easier to understand.

- P.Caponnetto's study; The way of writing (number and percentage) do not similarly other studies. If you want to express the same thing, please use the same expression and unify.

-> As requested, we wanted to write the results in the same format as other studies. Unfortunately, this paper only used standardized residual to present the results.

- Is "cigarette only" and "only cigarette users" same?

-> That is right. As requested, we have written, "cigarette-only users" instead of "only cigarette" to make it easier to understand.

- The P-value and 95% confidence interval are written differently for each paper, making it difficult to read.

-> As requested, we have written all in the same form for ease of reading.

Strength and limitation

- line286 It was written about vaccines. Why important the relationship of smoking and vaccination?

-> The following sentences and references have been added to the Discussion regarding smoking behavior that may change due to social distancing measures that change after vaccination on page 22, lines 288-293.

-> According to the results of this study, as social distancing and lockdown were strengthened in the early stages of the pandemic, smoking behavior, smoking cessation intention, and psychological changes appeared. However, in 2021, as vaccinations began to occur worldwide, social distancing decreased, as did people's willingness to participate in social distancing and stay-at-home. 5 Accordingly, smoking behavior, smoking cessation, and psychological status in 2021 may differ from 2020.

- There is almost no mention of the strengths of this research, but what are the strengths?

-> As requested, we have mentioned the strengths of this study on pages 21-22, lines 281-283 and 297-301.

-> This study is the first to systematically review the literature regarding changes in smoking behaviors, cessation intentions, and psychological states of smokers, based on the study period, country, and age of participants during the COVID-19 pandemic.

-> However, it is meaningful that this study looked at smoking usage behavior during the COVID-19 pandemic based on cigarette type. This study could serve as primary data to establish effective strategies that focus on specific tobacco-product users by suggesting different smoking behaviors, smoking cessation intentions, and psychological changes based on cigarette type.

Others

- Why did attach the yellow highlighted PDF file? What does it mean?

-> We marked the corrections in yellow highlights according to the comments of other reviewers.

Reviewer: 3

Miss Heike Garritsen, Amsterdam UMC Locatie AMC

Comments to the Author:

I thank the authors for taking into considerations my comments and altering the manuscript.

-> Thank you for your great comments and answer.

Reviewer: 2

Competing interests of Reviewer: None

Reviewer: 1

Competing interests of Reviewer: none

Reviewer: 3

Competing interests of Reviewer: None.

Editor(s)' Comments to Author (if any):

1. Hwang JH, Lyes M, Sladewski K, et al. Electronic cigarette inhalation alters innate immunity and airway cytokines while increasing the virulence of colonizing bacteria. *Journal of molecular medicine*. 2016;94(6):667-679.

2. Kaur G, Lungarella G, Rahman I. SARS-CoV-2 COVID-19 susceptibility and lung inflammatory storm by smoking and vaping. *Journal of Inflammation*. 2020/06/10 2020;17(1):21. doi:10.1186/s12950-020-00250-8

3. Chadi N, Schroeder R, Jensen JW, Levy S. Association between electronic cigarette use and marijuana use among adolescents and young adults: a systematic review and meta-analysis. *JAMA pediatrics*. 2019;173(10):e192574-e192574.

4. Chen Y, Klein SL, Garibaldi BT, et al. Aging in COVID-19: Vulnerability, immunity and intervention. Ageing research reviews. 2021;65:101205.

5. Andersson O, Campos-Mercade P, Meier AN, Wengström E. Anticipation of COVID-19 vaccines reduces willingness to socially distance. Journal of health economics. 2021;80:102530.

VERSION 3 – REVIEW

REVIEWER	Osaka International Cancer Institute Cancer Control Center, Department of Cancer Epidemiology
REVIEW RETURNED	26-Apr-2022

GENERAL COMMENTS	The manuscript is improved but is still riddled with some problems. The editor said "- Please complete a thorough proofread of the text and correct any spelling and grammar errors that you identify.". However, revised article included a miss spelling (for example; line 353 establish). Therefore, please confirm this matter one more. And, we can not read some sentence in Table1. Could you please fix up the appearance of Table 1?
--